# A life course perspective on predictors of midlife socioeconomic status

Erik Lykke Mortensen[1,2]*, Gunhild Tidemann Okholm[1,2,3], Trine Flensborg-Madsen[1,2], Merete Osler[3,4], Emilie Rune Hegelund[5]

1 Unit of Medical Psychology, Section of Environmental Health, Department of Public Health, University of Copenhagen, Copenhagen, Denmark, 2 National Institute of Public Health, University of Southern Denmark, Copenhagen K, Denmark, 3 Center for Clinical Research and Prevention, Copenhagen University Hospital – Bispebjerg and Frederiksberg, Frederiksberg, Denmark, 4 Section of Epidemiology, Department of Public Health, University of Copenhagen, Copenhagen, Denmark, 5 Methodology and Analysis, Statistics Denmark, Copenhagen, Denmark

* elme@sund.ku.dk

## Abstract

### Background

Previous studies have found paternal occupation, childhood intelligence, and educational attainment to be important predictors of socioeconomic status (SES) later in life. However, these factors only explain part of the variance in SES and thus, it is important to identify other predictors of SES and trajectories of influence from early childhood to adulthood.

### Objectives

To analyze predictors of SES attainment during the life course from early childhood to midlife with special emphasis on identifying direct and indirect effects on midlife SES of early childhood, late childhood and young adult characteristics.

### Methods

This study uses questionnaire and national registry data, including data on parental social background, intelligence, education, and midlife SES for 6,294 members of the Metropolit 1953 Danish Male Birth Cohort. The study sample included cohort members with information on intelligence at age 12 who were living in Denmark at age 50. Using structural equation modelling, direct and indirect mediated effects on midlife SES were estimated for early childhood, late childhood, and young adult characteristics.

### Results

Educational attainment, intelligence, parental education, and father's occupational class had the strongest influences on midlife SES. A prediction model only including

**Data availability statement:** The Metropolit database includes data from national registries and is consequently stored at Statistics Denmark. Data from Statistics Denmark are not allowed to be included in publicly available de-identified databases, and consequently our project database can only be accessed at Statistics Denmark. Researchers can get access to de-identified data through an institution approved by Statistics Denmark and with permission from the Metropolit steering group. Queries regarding data access and more information about the cohort can be found on the homepage: (https://www.frederiksberghospital.dk/ckff/sektioner/SBF/psykiatriskepidemiologi/Sider/The-Danish-Metropolit-cohort.aspx)

**Funding:** The author(s) received no specific funding for this work.

**Competing interests:** The authors have declared that no competing interests exist.

education and intelligence could only be slightly improved by the inclusion of other predictors, from 53.5% to 54.1% explained variance in midlife SES. Educational attainment was a particularly strong predictor of status attainment. Other early and late childhood factors had relatively weak direct effects, but significant indirect effects. Thus, it was possible to identify trajectories of influence from early childhood to midlife.

## Conclusion

Young adult education and intelligence were the strongest predictors of midlife SES. Early and late childhood factors influence young adult characteristics, but over the life course the direct effects of early life variables tend to decrease, and the effects on midlife SES become mediated and indirect.

## Introduction

Socioeconomic status (SES) has traditionally been defined to reflect individual differences in access to material and social resources [1]. Numerous studies have shown associations between SES and important life outcomes such as health and morbidity [2], and research has also shown associations of SES with cognition and emotions [3] as well as identity and self-concept [4]. This has prompted an interest in subjective social status [1] along with a wide interest in research on objective indicators of socioeconomic status.

Education, occupation and income have remained the most often used objective indicators of socioeconomic success or status [5]. However, there are important differences between education and the two other indicators of SES. Occupation and income can go up and down through the life course [6], while education is typically completed in young adulthood and, for this reason, can also be considered a predictor of later SES. In fact, parental SES, childhood intelligence, and educational attainment have been much investigated and may be considered established predictors of SES later in life [7,8]. The relative importance of parental SES, childhood intelligence, and educational attainment is difficult to establish because relatively few databases contain sufficient information on all three factors collected at relevant time points and because of methodological differences between studies.

In a meta-analytic review, Strenze compared parental characteristics, intelligence and academic performance (primarily grade point average) as predictors of socioeconomic success [6]. Several bivariate correlations between intelligence and the socioeconomic outcomes were larger, but not substantially larger, than the correlations for the two other predictors. Strenze [6] also analyzed changes in the predictive power of intelligence over time but did not find consistent evidence that intelligence has become a more important predictor of success in more recent cohorts, as suggested by Herrnstein and Murray [9]. However, this could still be the case for other predictors and it is likely that the relative influence of predictors of status attainment is context-dependent and consequently will differ both between countries and between

generations. For example, in Denmark, specific education and training became a requirement for many occupations in the second half of the 20th century. Consequently, education has likely become a more important predictor of socioeconomic attainment, and this may influence the relative importance of parental characteristics and intelligence. Thus, the importance of predictors of SES is likely to depend on context and culture and will to some extent vary between countries.

From a life course perspective, paternal occupational class may predict childhood intelligence and childhood intelligence may be an important factor in educational attainment. This suggests that effects on later life SES of parental characteristics could be indirect, mediated through effects on offspring intelligence and education, while effects of intelligence on SES could be mediated through effects on education. This was observed in a study which included measures of father's social class, childhood intelligence, education, and information on both young adult and midlife status attainment [10]. The study provided evidence of both direct and mediated indirect effects of all three factors on status attainment, suggesting that analyses that only evaluate direct effects on SES may underestimate the importance of early life factors. However, the study had a number of limitations such as retrospective information on important variables and a study sample consisting of only 240 Scottish men born in 1921.

While parental SES, intelligence and education together often explain substantial variance in later-life SES, several other potential predictors have also been investigated. These include physical characteristics such as birth weight [11] and adult height [12] as well as other family characteristics such as single-mother status [13] and number of siblings [14]. Furthermore, there has been increased interest in non-cognitive factors such as personality, life goals and interests [15,16]. Some of these factors (i.e., birth weight and height) tend to correlate with parental SES, intelligence and educational attainment, and it is important to estimate the independent and additional variance explained by these factors.

To illuminate these issues in a large Danish sample, the primary aim of this study was to provide a detailed analysis of predictors of SES attainment during the life course from childhood to midlife with special emphasis on estimating the relative importance of parental SES, intelligence and educational attainment and on evaluating the contribution of a broad range of other potential predictors of midlife SES.

## Methods

### Study design and participants

The Metropolit 1953 Danish Male Birth Cohort comprises all males born in the Greater Copenhagen Area in 1953 who survived until the age of 15 ($N = 11,532$) [17]. In 1965, 7,875 (68.3%) of the cohort members participated in a school-administered questionnaire survey, including an intelligence test. The main reasons for non-participation were emigration, lack of willingness of the school or class to participate in the survey, and absence from school on the test day. In 2003, when the cohort members were 50 years old, there were 6,294 men who had participated in the intelligence assessment at age 12 and were still alive and living in Denmark. These men were included in the study sample.

From the Metropolit database, we used data collected in birth registers and records, questionnaires and tests at ages 12–13 in schools, and national draft board examinations. Supplementary data were obtained from Statistics Denmark using the Danish civil registration numbers to link data from the Metropolit database and national registries.

The data analyses are structured according to the age of the cohort members at the time of data collection or the age at registration in national registries (see Fig 1). The included variables are described with midlife SES outcome variables first, followed by young adult, late childhood and early childhood predictors to enable an overview of the data structure and analyses.

### Measures

**Outcome: Midlife socioeconomic status (SES).** Midlife SES was measured by a latent variable using information on the individual's occupational class and income at age 50. There is some evidence that associations of intelligence with SES are stronger in midlife than in young adulthood [6,18]. Age 50 for assessing SES was chosen because the majority

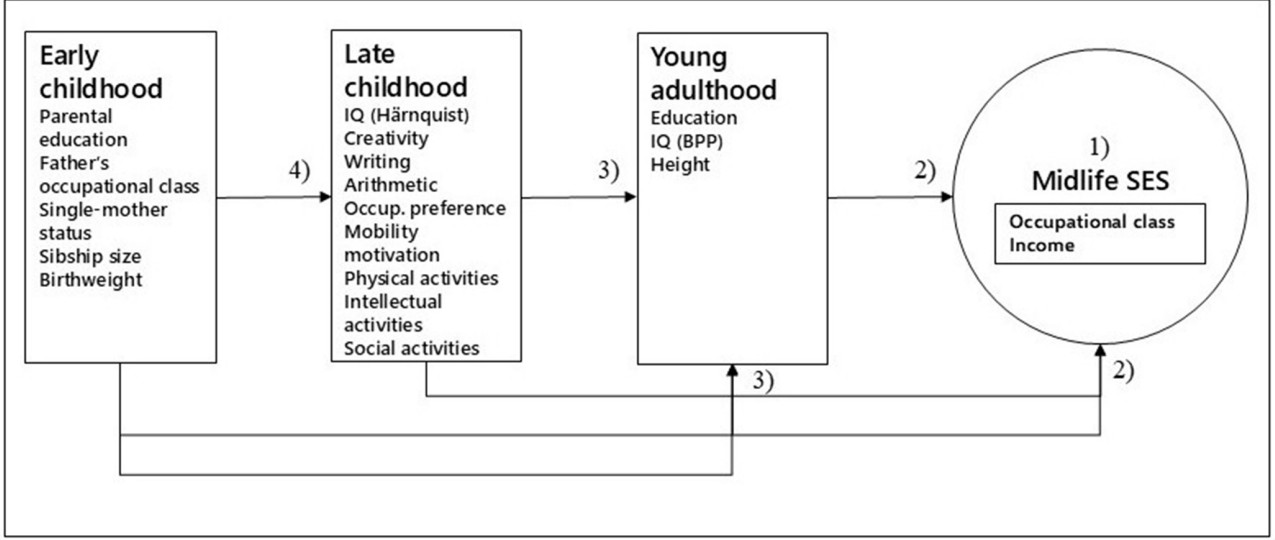

Outline of the midlife SES latent variable, the early childhood, late childhood, and young adulthood predictors of midlife SES, and the model components included in the analyses of direct and indirect influences of early predictors on midlife SES. Numbers refer to the analyses described in the data analysis section and in tables 2-4.

**Fig 1. Overview of the included predictors of midlife SES and the two indicators of midlife SES.**

of the cohort members were expected to have reached their highest occupational class, and the occupational class and income of relatively few would be affected by health problems or other age-related factors.

The individual's occupational class was categorized into five categories in accordance with previous Danish studies [19]. *Category 5*: University graduates **or** top-level management positions. *Category 4*: Jobs requiring medium-long education **or** management responsibilities for 11–50 subordinates. *Category 3:* Jobs requiring short theoretical education **or** management responsibilities for up to 10 subordinates. *Category 2:* Lower level white-collar jobs and skilled manual workers. *Category 1*: Unskilled workers. Self-employment was also coded according to the number of subordinates, but for each category required fewer subordinates than salaried employment.

Income was indicated by the family's equivalised disposable income and was winzorised at the 1st and 99th percentile. Information on both variables was obtained from Statistics Denmark.

For the measurement model, the standardized loadings were 0.89 for occupational class and 0.45 for income. The higher loading on occupational class partly reflects the fact that this variable tended to have higher correlations with the main predictor variables than the income variable.

**Young adult predictors.** *Education (age 30):* Education was measured as the individual's highest educational level at age 30 and categorized into 8 categories according to the International Standard Classification of Education. This classification ranks educational levels from category 1 (primary education) to category 8 (doctoral or equivalent level). Category 4 (post-secondary non-tertiary education) is, however, not used in Denmark, and consequently, educational levels were recoded to a 1–7 point scale in the statistical analyses. Information on education was obtained from Statistics Denmark. *Young adult intelligence score (typically age 18–21):* All Danish men must appear before a draft board at the age of 18. The draft board examination includes the administration of the BPP (Børge Priens Prøve), which is the Danish military

intelligence test [20]. The test comprises four subtests: letter matrices (19 items), verbal analogies (24 items), number series (17 items), and geometric figures (18 items). The score is the total number of correct answers (range 0–78), but in this study, the total score was transformed into an IQ scale with a sample mean of 100 and a standard deviation of 15. The BPP total score has been shown to correlate 0.82 with the Wechsler Adult Intelligence Scale [21]. The BPP scores were obtained from national draft board registries.

*Young adult height (typically age 18–21):* The draft board examination includes measurement of height measured without shoes and to the nearest centimeter, which is recorded together with the intelligence test scores.

**Late childhood predictors.** *Late childhood intelligence score (age 12):* Late childhood intelligence was measured by the Härnqvist intelligence test [22]. This battery consists of three subtests: verbal analogies, number series, and geometric figures. Each of the three subtests includes 40 items, and the number of correct answers is summed to a total score (range: 0–120), but in the current study, the total score was transformed into an IQ scale with a sample mean of 100 and a standard deviation of 15. The total Härnqvist score has previously been found to correlate 0.69 with the BPP intelligence test administered by the draft board [23].

*Creativity (age 12):* A 25-item version of Mednick's Remote Associates Test (RAT) was completed at the 12-year follow-up. The RAT is based on Mednick's theory of creativity [24]. The test requires the subject to find a word that connects three other words without obvious connections, and the score is assumed to reflect creativity or divergent thinking [25].

*Leisure time activities (age 12):* A previous study developed a measure based on 15 items on leisure activities included in a questionnaire [23]. Three categories of activities were scored on a 0–5 point scale: Physical activities (football, handball, sailing, swimming, and gym), intellectual activities (literature, technical books, books on animals, history books, and physics and chemistry books), and social activities (be with friends at home, meet with friends, be with family, evening school, and youth club).

*Achievement motivation (age 12):* The cohort members were asked to indicate their preferred future occupation among 52 different jobs. The occupational class of the respondent's choice was coded on a 1–5 point scale. The participants were also asked in a yes-no response format whether they wanted to reach a higher social position than their father. The two variables were called occupation preference and social mobility motivation.

*Academic performance (ages 12–13):* Approximately one year after administration of the intelligence test, results of a Danish writing test and an arithmetic test, routinely administered by schools, were recorded for the cohort members. The Danish writing test was scored on a 0–5 point scale and the arithmetic test on a 0–9 point scale with higher scores indicating better academic performance.

**Early childhood predictors.** *Parental education attainment (register-based):* This variable was coded as the level of education for the parent with the longest education if information was available for both parents. It was coded for one parent if this was the only information, and it was coded as missing if information was unavailable for both parents. Parental education was also coded according to The International Standard Classification of Education, and the information was obtained from Statistics Denmark (for the majority the registration was from 1970, but it was assumed that most parents would have completed their education in 1953).

*Father's occupational class (at birth):* Father's occupation at birth was coded into 23 categories, and this information was categorized into five ordered categories of occupational class in accordance with the midlife occupational class variable used for the participants to derive the midlife SES variable. This information was registered in the Metropolit database but was originally transferred from birth registers and certificates [26]. If the information on the father's occupation was insufficient, it was supplemented with information on paternal education to derive the coding of occupational class. Mother's occupational class was not registered because the great majority of women were housewives in 1953.

*Single mother status (at birth):* A binary indicator variable was used to code the mother's marital status at birth. This information was registered in the Metropolit database but was originally transferred from birth registers and certificates [26].

*Sibship size (age 12):* This variable coded the number of siblings born before the index member of the Metropolit cohort was 12 years old. The information was obtained from Statistics Denmark.

*Birth weight (at birth):* This information was registered in the Metropolit database but was originally transferred from birth registers and certificates [26].

## Data analysis

Stata's sem procedure was used for the detailed analysis of the Metropolit data in one comprehensive model, which included one latent variable (midlife SES) and a number of regression models with observed variables as outcomes. The comprehensive model included midlife SES, young adult characteristics, as well as late and early childhood characteristics. To track the direct and indirect influences of early predictors over the life course, the model included the following components illustrated in Fig 1 (the numbers in the figure refer to the four model components below):

1) A measurement model defining midlife SES as a latent variable and including midlife occupational status and income as observed variables

2) A regression model regressing midlife SES on young adult characteristics, as well as late childhood and early childhood characteristics (Table 2)

3) Regression models regressing young adult education, intelligence and height on late and early childhood characteristics (Table 3)

4) Regression models regressing all late childhood characteristics on early childhood characteristics (Table 4)

Late childhood intelligence was not included in the midlife SES regression model (2) because of the substantial correlation with young adult intelligence. Late childhood leisure activities and social mobility motivation were also excluded because they were considered relatively unstable and less important as long-term predictors (this was confirmed in preliminary analyses).

To avoid diluting the sample because of missing data, Stata's procedure for full information maximum likelihood analyses (FIML) was used. The associations are presented using standardized regression coefficients to make it easier to compare the relative importance of the predictors in each model.

In all regression models, there was a time interval between the registration of predictors and outcome and between predictors registered at different ages, making it meaningful to use Stata's facilities to evaluate direct and indirect mediated effects. Thus, the effects of early childhood characteristics on midlife SES could be indirect, mediated through effects on late childhood and young adult characteristics, while effects of late childhood characteristics could be mediated through effects on young adult characteristics. These pathways are shown in Fig 2.

Stata's modification indices suggested that a number of error covariances should be allowed to obtain a sufficiently good model fit. Thus, error covariances were allowed if modification indices were larger than 100. Most of the involved variables were late childhood characteristics, which may all have been influenced by the child's development at this stage, or they were closely related with respect to content or data collection methods.

For the model including the error covariances, RMSEA was 0.042 and CFI 0.979.

## Ethics

According to the Danish law on scientific ethical committees, all health science projects must be reported to and approved by the local scientific ethics committee (§ 14 in the law). However, the law specifies several exceptions to this general rule, such as 14.2, which states that projects based on questionnaires and/or register data should only be reported and approved if the project also involves human biological material. Since our analyses do not involve biological material, this study does not require approval by the Danish scientific ethical committee system. However,

In addition to direct effects on midlife SES, effects of early childhood may be mediated through late childhood and young adult predictors. In addition to direct effects, effects of late childhood may be mediated through young adult predictors.

**Fig 2. Indirect pathways between early and late childhood and midlife SES.**

register-based databases must be approved by the Danish Data Protection Agency. Originally, the Metropolit Cohort database was approved by the Danish Data Protection Agency in 2001and when the authority to approve was delegated to the universities, University of Copenhagen approved the database in 2018. The data were assessed for the current analyses on May 29, 2024.

Since this is a register-based study without contact with the participants, informed consent was not obtained for our analyses. This is in accordance with the Danish rules for registry based studies, but in addition several factors make informed consent problematic, including the fact that many parents and Metropolit cohort members were dead when the analyses were conducted in 2024.

All Danes are assigned a unique 10-digit CPR number, which Statistics Denmark uses to link information from different registries, but the CPR number is not included in research databases. This means that research databases are pseudo anonymized in the sense that the researcher cannot identify the individuals in the database, but Statistics Denmark will be able to link to the CPR numbers, which is necessary to able to link the database to other registries.

## Results

### Descriptive data

Sample means, SDs, and selected correlations are presented in Table 1. The table presents correlations for the two indicators of midlife SES and the previously well-established predictors parental education, father's occupational class, young adult IQ, and own education. Both the midlife SES indicator variables showed relatively high correlations with these four predictors, but the correlations were systematically higher for midlife occupational class than for income. Among the young adult predictors, education showed the strongest correlations with the two SES indicators, but the correlations were also high for young adult IQ.

Table 1 also showed relatively high inter-correlations among the four established predictors with the highest inter-correlations for the two parental characteristics (0.54) and the two young adult characteristics (0.44). Among the late childhood variables, one group was clearly related to young adult intelligence and education, but also to the two SES indicators, as well as parental education and father's occupational class.

**Table 1. Sample means, SDs, and selected correlations.**

| Variables | Basic Statistics | | | Established Predictors Correlations | | | | SES Indicators Correlations | |
|---|---|---|---|---|---|---|---|---|---|
| | N | Mean | SD | Par. educ. | Fatheroccup. | BPP IQ | Y. A. Educ. | Occup. | In-come |
| **Midlife** | | | | | | | | | |
| Occupational class (1–5 point scale) | 6292 | 2.64 | 1.40 | 0.26 | 0.27 | 0.43 | 0.63 | --- | 0.40 |
| Income | 6248 | 12.20 | 0.40 | 0.11 | 0.11 | 0.23 | 0.31 | 0.40 | --- |
| **Young adulthood** | | | | | | | | | |
| Young adult education (ISCED level) | 6290 | 3.34 | 1.16 | 0.31 | 0.30 | 0.44 | --- | 0.66 | 0.31 |
| Young adult IQ (BPP) | 5845 | 100.00 | 15.00 | 0.30 | 0.32 | --- | 0.44 | 0.43 | 0.23 |
| Height (cm) | 5852 | 178.47 | 6.55 | 0.11 | 0.13 | 0.20 | 0.18 | 0.19 | 0.12 |
| **Late childhood** | | | | | | | | | |
| Childhood IQ (Härnquist test) | 6294 | 100.00 | 15.00 | 0.27 | 0.28 | 0.73 | 0.36 | 0.35 | 0.18 |
| Mednick creativity test | 6238 | 8.96 | 5.26 | 0.19 | 0.20 | 0.48 | 0.29 | 0.27 | 0.13 |
| Danish writing test (0–5 point scale) | 5455 | 2.48 | 1.29 | 0.21 | 0.23 | 0.45 | 0.29 | 0.27 | 0.13 |
| Arithmetic test (0–10 point scale) | 5465 | 5.67 | 1.21 | 0.12 | 0.14 | 0.40 | 0.22 | 0.23 | 0.15 |
| Occupation preference (1–5 point scale) | 5965 | 3.27 | 1.23 | 0.31 | 0.33 | 0.38 | 0.28 | 0.25 | 0.09 |
| Social mobility motivation (binary scale) | 5865 | 0.41 | 0.49 | −0.08 | −0.12 | 0.06 | 0.02 | 0.03 | 0.00 |
| Physical activities (0–5 point scale) | 5990 | 2.62 | 1.43 | −0.08 | −0.08 | −0.11 | −0.06 | −0.05 | −0.01 |
| Intellectual activities (0–5 point scale) | 5841 | 1.84 | 1.37 | 0.08 | 0.07 | 0.11 | 0.10 | 0.07 | 0.01 |
| Social activities (0–5 point scale) | 5899 | 2.36 | 1.44 | −0.01 | 0.00 | −0.06 | 0.00 | −0.01 | −0.01 |
| **Early childhood** | | | | | | | | | |
| Parental education (ISCED level) | 5026 | 2.94 | 1.12 | --- | 0.54 | 0.30 | 0.31 | 0.26 | 0.11 |
| Father's occupational class (1–5 point scale) | 6280 | 2.54 | 1.25 | 0.54 | --- | 0.32 | 0.30 | 0.27 | 0.11 |
| Single-mother status (binary scale) | 6114 | 0.06 | 0.25 | −0.07 | −0.08 | −0.09 | −0.09 | −0.10 | −0.07 |
| Sibship size | 6076 | 0.87 | 0.93 | 0.00 | −0.05 | −0.04 | −0.06 | −0.05 | −0.01 |
| Birth weight (kg) | 6114 | 3.39 | 0.53 | 0.01 | 0.02 | 0.05 | 0.02 | 0.03 | 0.05 |

Correlations at 0.03 or larger are statistically significant.

*The two occupational class variables are based on different information and the means are not directly comparable.

This group comprised childhood IQ, the creativity test, the two school performance tests and occupation preference. Notably, these variables may still independently explain variance in midlife SES since the correlations with the established predictors suggest limited overlapping variance.

In contrast, late childhood social mobility motivation and activities as well as the early childhood variables single-mother status, sibship size and birth weight generally showed weak correlations with the established predictors and the two SES indicators (in particular income). In spite of the apparent weak correlations with midlife SES, these variables may contribute to prediction because of very limited overlapping variance with other predictors in the regression models.

Height showed somewhat higher correlations with young adult intelligence, education and occupational social class, but these correlations were clearly lower than the correlations of late childhood IQ and education-related characteristics with the corresponding young adult and midlife characteristics.

### Direct effects on midlife SES

The primary direct predictor of midlife SES was educational attainment at age 30, followed by young adult IQ (Table 2). The other significant predictors were young adult height, late childhood creativity and arithmetic test scores, and the early childhood factors father's occupational class and being born by a single mother. The lack of significant direct effects for

parental education may partly reflect the sample correlation between parental education and father's occupational class, which was 0.54.

Overall, the explained variance for midlife SES was 54.1% for the included variables, but a supplementary analysis showed that a model only including young adult IQ and education explained 53.5% of the variance (adding parental education and father's occupational class did not increase this proportion). Thus, the explained variance in midlife SES only increased by 0.6% when including the remaining 10 young adult and childhood variables, and further analysis showed that it only increased slightly (0.12%) if childhood IQ and the other excluded childhood variables were included in the prediction of midlife SES.

### Indirect effects on midlife SES

Several late and early childhood variables showed significant indirect effects on midlife SES. Among late childhood variables, IQ showed the strongest indirect effects, followed by the academic performance tests and occupation preference. These mediated effects reflect direct effects on young adult characteristics. Thus, Table 3 shows that these variables all significantly predicted young adult education and IQ, and all except the arithmetic test predicted young adult height. Indirect effects on midlife SES of early childhood variables may reflect direct effects on late childhood characteristics (Table 4), as well as direct and indirect effects on young adult characteristics (Table 3). Parental education and father's occupational class showed the most substantial indirect effects on midlife SES, reflecting significant direct and indirect effects on young adult characteristics, and substantial direct effects of parental education and father's occupational class

**Table 2. Standardized direct and indirect effects on midlife socioeconomic status**.**

|  | Direct effects | Indirect effects | Total effects |
|---|---|---|---|
| **Explained variance** | **54.1%** |  |  |
| **Young Adult Predictors** |  |  |  |
| Education | 0.59*** | – | 0.59*** |
| IQ (BPP) | 0.16*** | – | 0.16*** |
| Height | 0.06*** |  | 0.06*** |
| **Late Childhood Predictors** |  |  |  |
| IQ (Härnquist) | – | 0.19*** | 0.19*** |
| Creativity | 0.03* | 0.03** | 0.06*** |
| Danish writing test | −0.01 | 0.06*** | 0.05** |
| Arithmetic test | 0.04** | 0.05*** | 0.09*** |
| Occupation preference | 0.01 | 0.06*** | 0.07*** |
| Social mobility motivation | – | 0.01 | 0.01 |
| Physical activities | – | −0.01 | −0.01 |
| Intellectual activities | – | 0.03** | 0.03** |
| Social activities | – | −0.01 | −0.01 |
| **Early Childhood Predictors** |  |  |  |
| Parental education | 0.02 | 0.16*** | 0.19*** |
| Father's occupational class | 0.03* | 0.16***. | 0.19*** |
| Single-mother status | −0.04** | −0.05*** | −0.08*** |
| Sibship size | −0.01 | −0.04*** | −0.05** |
| Birth weight | −0.00 | 0.03*** | 0.03* |

*$p < 0.05$; **$p < 0.01$; ***$p < 0.001$.

**The variables without a coefficient for direct effects were not included in the regression model predicting midlife socioeconomic status.

**Table 3. Standardized direct and indirect effects on young adult education, intelligence, and height.**

|  | Direct effects | Indirect effects | Total effects |
|---|---|---|---|
| **Young Adult Education** | **Expl. var = 21.3%** |  |  |
| **Late Childhood Predictors** |  |  |  |
| IQ (Härnquist) | 0.16*** | – | 0.16*** |
| Creativity | 0.05** | – | 0.05** |
| Danish writing test | 0.07*** | – | 0.07*** |
| Arithmetic test | 0.06*** | – | 0.06*** |
| Occupation preference | 0.07**** | – | 0.07*** |
| Social mobility motivation | 0.02 | – | 0.02 |
| Physical activities | −0.02 | – | −0.02 |
| Intellectual activities | 0.04** | – | 0.04** |
| Social activities | −0.00 | – | −0.00 |
| **Early Childhood Predictors** |  |  |  |
| Parental education | 0.15*** | 0.06*** | 0.21*** |
| Father's occupational class | 0.11*** | 0.07*** | 0.18*** |
| Single-mother status | −0.04** | −0.02*** | −0.06*** |
| Sibship size | −0.04** | −0.01** | −0.05*** |
| Birth weight | 0.00 | 0.02*** | 0.02 |
|  |  |  |  |
| **Young Adult Intelligence – BPP** | **Expl. var = 58.4%** |  |  |
| **Late Childhood Predictors** |  |  |  |
| IQ (Härnquist) | 0.58*** | – | 0.58*** |
| Creativity | 0.02 | – | 0.02 |
| Danish writing test | 0.10*** | – | 0.10*** |
| Arithmetic test | 0.10*** | – | 0.10*** |
| Occupation preference | 0.08*** | – | 0.08*** |
| Social mobility motivation | 0.03** | – | 0.03** |
| Physical activities | −0.02 | – | −0.02 |
| Intellectual activities | 0.02* | – | 0.02* |
| Social activities | −0.03** | – | −0.03** |
| **Early Childhood Predictors** |  |  |  |
| Parental education | 0.03* | 0.13*** | 0.17*** |
| Father's occupational class | 0.08*** | 0.15*** | 0.23*** |
| Single-mother status | −0.01 | −0.04*** | −0.05*** |
| Sibship size | −0.01 | −0.02* | −0.03* |
| Birth weight | 0.00 | 0.04*** | 0.05*** |
|  |  |  |  |
| **Young Adult Height** | **Expl. var = 9.9%** |  |  |
| **Late Childhood Predictors** |  |  |  |
| IQ (Härnquist) | 0.09*** | – | 0.09*** |
| Creativity | −0.01 | – | −0.01 |
| Danish writing test | 0.04** | – | 0.04** |
| Arithmetic test | 0.02 | – | 0.02 |
| Occupation preference | 0.05*** | – | 0.05*** |
| Social mobility motivation | 0.01 | – | 0.01 |
| Physical activities | −0.02 | – | −0.02 |

*(Continued)*

**Table 3.** (Continued)

| | Direct effects | Indirect effects | Total effects |
|---|---|---|---|
| Intellectual activities | 0.02 | – | 0.02 |
| Social activities | 0.01 | – | 0.01 |
| **Early Childhood Predictors** | | | |
| Parental education | 0.03 | 0.03*** | 0.06*** |
| Father's occupational class | 0.05* | 0.04*** | 0.09*** |
| Single-mother status | −0.00 | −0.01*** | −0.01 |
| Sibship size | −0.04** | −0.01*** | −0.05*** |
| Birth weight | 0.23*** | 0.01** | 0.24*** |

*$p < 0.05$; **$p < 0.01$; ***$p < 0.001$.

**Table 4. Standardized direct effects on late childhood characteristics predicted by early childhood characteristics.**

| Late Childhood Characteristics | R squared | Parental education | Father's occupational class | Single-mother status | Sibship size | Birth weight |
|---|---|---|---|---|---|---|
| IQ (Härnquist) | 0.10 | 0.16*** | 0.18*** | −0.06*** | −0.02 | 0.06*** |
| Creativity | 0.05 | 0.11*** | 0.13*** | −0.01 | −0.04** | 0.04** |
| Danish writing test | 0.07 | 0.14*** | 0.16*** | −0.03* | −0.00 | 0.03* |
| Arithmetic test | 0.03 | 0.07*** | 0.10*** | −0.04** | −0.03* | 0.06*** |
| Occupation preference | 0.14 | 0.18*** | 0.23*** | −0.00 | −0.05*** | −0.02 |
| Social mobility motivation | 0.02 | −0.01 | −0.12*** | −0.02 | −0.03* | −0.00 |
| Physical activities | 0.01 | −0.06** | −0.04** | 0.03* | 0.04** | −0.01 |
| Intellectual activities | 0.01 | 0.06** | 0.04** | 0.01 | −0.01 | −0.02 |
| Social activities | 0.00 | −0.02 | 0.02 | 0.02 | 0.01 | −0.02 |

*$p < 0.05$; **$p < 0.01$; ***$p < 0.001$.

on late childhood IQ and related variables. Single-mother status, sibship size and birth weight all showed significant indirect effects on midlife SES. These mediated effects reflected significant direct effects on young adult and late childhood characteristics (see Tables 3 and 4).

## Total effects on midlife SES

The only variables without significant total effects on midlife SES were physical and social activities in childhood. Thus, educational attainment was the variable with the most substantial total effect, followed by childhood IQ, parental education, father's occupational class, and young adult IQ. Less strong predictors were creativity, academic performance, occupation preference, intellectual activities, birth weight, and young adult height.

## Discussion

We used the Metropolit cohort comprising Danish men born in 1953 to analyze associations of predictors of SES attainment in midlife. Our model not only analyzed predictors of midlife status attainment but also predictors of young adult characteristics and late childhood characteristics, enabling analyses of indirect or mediated effects of early life factors. While our findings corroborate previous studies identifying education, intelligence and parental characteristics as important predictors of SES, the analyses also showed that several other early and later characteristics had direct or indirect effects on midlife SES. On the one hand, this means that studies only estimating direct effects in regression models will

tend to underestimate the developmental importance of early life variables. On the other hand, supplementary analyses showed that a model including only young adult education and intelligence only explained slightly less variance than the model with early life predictors, which underlines the importance of young adult characteristics as predictors of midlife SES.

It is important to note that the regression models were based on the available data in the Metropolit cohort. Thus the models do not necessarily include all relevant predictors and this will of course influence the estimates (for example, a measure of late childhood height was not available). Consequently, the observed direct and indirect effects do not necessarily reflect causal effects but describe the pattern of associations among the available variables.

## Comparison with previous studies

Parental characteristics are among the established predictors of offspring SES [6,7], but compared to the existing literature, father's occupational class and parental education had relatively weak direct effects on midlife SES in the present study. In fact, the indirect effects of these two variables were much larger, reflecting their strong associations with young adult education and with late childhood IQ and related variables. These associations explain the strong indirect effects on midlife SES but also suggest a decreasing role of factors directly associated with parental characteristics in midlife. In fact, the direct effects of parental education and father's occupational class on late childhood variables related to intelligence were relatively strong, weaker on young adult education and intelligence, and weakest on midlife SES. A somewhat similar pattern was observed for birth weight, which had direct effects on late childhood intelligence and related variables but no direct effects on midlife SES or young adult education and intelligence. Similarly, sibship size showed a significant negative direct effect on education but no significant direct effect on midlife SES. However, there are studies showing associations between birth weight and midlife intelligence [27] and between the number of younger siblings and midlife income when IQ and education are included as covariates [14].

In contrast, single-mother status shows similar direct effects on young adult education and midlife SES, raising the issue of the nature of the effects of single-mother status. In the sample, about 6% were offspring of single mothers, but it is likely that many mothers only stayed single for a limited period, and in this perspective, it is remarkable that single-mother status at birth was a significant predictor of midlife SES. A previous Danish study has observed a significant long-term effect of single-mother status at birth on offspring intelligence in midlife [28], but in our study, single-mother status showed no direct effect on young adult intelligence, and young adult intelligence was included in our prediction model.

The late childhood variables related to intelligence and education (creativity, school performance and occupation preference) showed a pattern of relatively strong direct effects on young adult education but weaker effects on midlife SES with only the creativity and arithmetic tests showing significant coefficients. Educational attainment and midlife SES are not identical outcomes, but the patterns for several variables suggest that the direct effects of early life predictors tend to become diluted as the individual develops and enters new life phases. However, because early life variables predicted young adult characteristics, all early childhood variables and most late childhood variables had significant indirect effects on midlife SES, and consequently also significant total effects on midlife SES attainment. The total effects in Table 2 clearly show that the variables with the strongest influences on midlife SES are the established predictors education, intelligence, parental education, and father's occupational class.

Young adult height had the third strongest direct effect on midlife SES after young adult education and IQ. Height correlates with intelligence, education, parental education, and father's occupational class, but since these variables were all included in the prediction of midlife SES, the issue here is the consequences during adult life of individual differences in young adult height. As our outcome was midlife SES, associations between height and morbidity may potentially play a role, but the relationship between height and morbidity is complex [12], and perhaps psychosocial factors – such as social advantages of being tall – are also important [12]. Notably, there was a strong direct effect of birth weight on young adult height, corroborating previous evidence that adult height partly mediates associations between birth weight and midlife income [11].

## The importance of education

Young adult education was clearly the strongest predictor of midlife SES, substantially stronger than intelligence. Educational attainment may be a strong predictor not only because tertiary education provides specific skills required for certain occupations, but also because it is an important indicator of non-cognitive skills related to success in many occupations, including personality factors such as conscientiousness and emotional stability [15,16]. Compared to the Danish results, late childhood intelligence and father's occupational class played a relatively larger role in the Scottish sample [10]. These differences could both reflect generational (the Scottish sample was born in 1921) and cultural differences between the two countries, as education may be more important in younger generations, and father's occupational class may have been more important in older generations and in a country characterized by more substantial class differences.

The Danish men born in 1953 entered the labor market in the 1970s. For this generation of Danish men, there was a very strong link of education with occupation and income, as the educational system provided specific qualifications for specific jobs (Danish university studies were designed to qualify students for specific civil service positions). In contrast, individual characteristics (late childhood intelligence) and social background (father's occupational class) seemed to play a relatively larger role among Scottish men, perhaps to some extent reflecting a weaker link between education and type of job. However, a study of Scottish men born in 1950−56 observed a stronger direct effect of education than of intelligence but found intelligence to be the strongest predictor of education [8]. These results corroborate our results and provide evidence of generation differences in Scotland. Furthermore, a recent Danish study has observed increasing correlations of education with income and employment for younger generations of Danes [29].

## Strengths and limitations

Among the strengths of this study is national registry information on crucial variables, such as midlife occupation, income and education, and the prospective registration of Metropolit data. It is also a strength that all predictor variables in the regression models were recorded several years before the predicted outcomes, making the estimation of direct and indirect effects meaningful, although the statistical estimates do not necessarily reflect causal effects. Finally, the large sample size, providing sufficient statistical power, should also be considered a strength of the study.

There are some limitations in our midlife SES measure. First, many would consider a formative model more appropriate if SES is conceived as a combination of occupational class and income. However, our analyses were based on a reflective measurement model because we were unable to fit a formative model, as it did not converge. Second, it was based on occupational class and income, and our bivariate correlations are in line with evidence that it is difficult to predict income [6]. However, the latent variable primarily reflected occupational class, and the regression model explained about 54% of the variance in our midlife SES latent variable.

We used the family's equivalised disposable income but considered using a measure of disposable personal income. However, the family's equivalised disposable income was chosen because it is a more comprehensive measure of SES, taking into account all family members' income and adjusting for the number of adults and children in the family. Furthermore, this measure tended to show higher correlations with the included predictor variables. The correlation between the two income variables was 0.77, and a model using personal income showed very similar results to the model based on the family's equivalised disposable income.

We did not analyze interactions between predictors (e.g., early childhood factors and education), as this would require a separate paper. We were also unable to investigate neighborhood effects possibly related to parental SES, although such factors may play an increasing role in Denmark [30].

The Metropolit cohort was established to analyze social mobility, and as a consequence, the database includes many relevant variables for analyzing status attainment. However, some of the information was only available for part of the sample, and at age 50, mortality may play a role. Furthermore, it is an important weakness that a standard measure of personality was not included, limiting analyses of the influence of non-cognitive factors on status attainment. Finally,

because of the focus on social mobility and because few women were in the labor market when the Metropolit cohort was established, only men were included in the cohort, precluding generalizations of results to women.

Future studies should strive to incorporate genetic information, which may contribute to understanding the complex relationships among correlated factors influencing development. Thus, a genome-wide polygenic score (GPS) predicting educational achievement has been found to be associated with a wide range of life-course outcomes, including adult economic outcomes [31].

## Conclusion

In conclusion, our study demonstrates that the established predictors educational attainment, intelligence, parental education and father's occupational class had the strongest influence on midlife SES. However, a prediction model only including young adult education and intelligence could only be slightly improved by the inclusion of other predictors, and educational attainment was a particularly strong predictor of status attainment for this generation of Danes. While we identified a relatively wide range of relevant early-life variables, only a few showed direct effects on midlife SES and the effects were generally small. However, with the available data for the Metropolit cohort, it was possible to identify trajectories of influence from early childhood to late childhood, young adulthood and later midlife. Over the life course, the direct effects of early childhood and late childhood factors tend to decrease, but the mediated indirect effects tend to become important, underlining the importance of early life factors for young adult development and midlife SES. Comparison with the existing literature underlines the importance of generation differences, as well as historical and cultural context. Consequently, it remains an open question whether results for Danish men born in 1953 can be generalized to younger generations of Danish men and women.

## Author contributions

**Conceptualization:** Erik Lykke Mortensen, Merete Osler, Emilie Rune Hegelund.

**Data curation:** Merete Osler, Emilie Rune Hegelund.

**Formal analysis:** Emilie Rune Hegelund.

**Investigation:** Erik Lykke Mortensen, Gunhild Tidemann Okholm, Trine Flensborg-Madsen, Merete Osler, Emilie Rune Hegelund.

**Methodology:** Erik Lykke Mortensen, Gunhild Tidemann Okholm, Trine Flensborg-Madsen, Merete Osler, Emilie Rune Hegelund.

**Project administration:** Erik Lykke Mortensen, Merete Osler, Emilie Rune Hegelund.

**Writing – original draft:** Erik Lykke Mortensen.

**Writing – review & editing:** Erik Lykke Mortensen, Gunhild Tidemann Okholm, Trine Flensborg-Madsen, Merete Osler, Emilie Rune Hegelund.

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
