## [Decision Letter · Decision Letter 0]

28 May 2025

PONE-D-25-14004A life course perspective on predictors of midlife socioeconomic statusPLOS ONE

Dear Dr. Mortensen,

Thank you for submitting your manuscript to PLOS ONE. After careful consideration, we feel that it has merit but does not fully meet PLOS ONE’s publication criteria as it currently stands. Therefore, we invite you to submit a revised version of the manuscript that addresses the points raised during the review process.

We look forward to receiving your revised manuscript.

Kind regards,

Heather Macdonald, Ph.D

Academic Editor

PLOS ONE

**Journal Requirements:**

1. When submitting your revision, we need you to address these additional requirements. Please ensure that your manuscript meets PLOS ONE's style requirements, including those for file naming. The PLOS ONE style templates can be found at https://journals.plos.org/plosone/s/file?id=wjVg/PLOSOne_formatting_sample_main_body.pdf and https://journals.plos.org/plosone/s/file?id=ba62/PLOSOne_formatting_sample_title_authors_affiliations.pdf 2. We note that you have indicated that there are restrictions to data sharing for this study. For studies involving human research participant data or other sensitive data, we encourage authors to share de-identified or anonymized data. However, when data cannot be publicly shared for ethical reasons, we allow authors to make their data sets available upon request. For information on unacceptable data access restrictions, please see http://journals.plos.org/plosone/s/data-availability#loc-unacceptable-data-access-restrictions.  Before we proceed with your manuscript, please address the following prompts: a) If there are ethical or legal restrictions on sharing a de-identified data set, please explain them in detail (e.g., data contain potentially identifying or sensitive patient information, data are owned by a third-party organization, etc.) and who has imposed them (e.g., a Research Ethics Committee or Institutional Review Board, etc.). Please also provide contact information for a data access committee, ethics committee, or other institutional body to which data requests may be sent. b) If there are no restrictions, please upload the minimal anonymized data set necessary to replicate your study findings to a stable, public repository and provide us with the relevant URLs, DOIs, or accession numbers. Please see http://www.bmj.com/content/340/bmj.c181.long for guidelines on how to de-identify and prepare clinical data for publication. For a list of recommended repositories, please see https://journals.plos.org/plosone/s/recommended-repositories. You also have the option of uploading the data as Supporting Information files, but we would recommend depositing data directly to a data repository if possible. Please update your Data Availability statement in the submission form accordingly. 3. Please amend either the abstract on the online submission form (via Edit Submission) or the abstract in the manuscript so that they are identical.

**Additional Editor Comments:**

In addition to the comments from the 2 reviewers, please consider the following:

1. Please review the PLOS One guidelines for statistical reporting and ensure that the manuscript complies (e.g., exact p values when greater than 0.001, confidence intervals etc): https://www.equator-network.org/wp-content/uploads/2013/03/SAMPL-Guidelines-3-13-13.pdf

2. Please also consider guidelines for SEM analysis and ensure that all relevant details are included in the manuscript (e.g., https://doi.org/10.1016/j.anr.2021.06.001). It may help to review how other SEM-based papers in PLOS One have reported their findings. It seems that a path analysis diagram showing the direct and indirect effects is warranted, and may also help illustrate the "trajectories of influence" that the authors mention in their conclusion. In general, for a non-SEM expert, I found it difficult to follow the results and in their current form, some of the tables cannot stand alone from the text as it isn't clear what values are presented.

3. Please provide the rationale for only including father's occupational class.

4. Page 11, Line 16: What preliminary analyses were conducted and where are these reported?

5. Table 1: In the footnote, the authors mention 3 occupational class variables but I only see 2 listed in the table.

6. Page 5, Line 22: Consider clarifying that you are referring to adult height in this sentence.

7. Page 21, Lines 22-24: Regarding the statement that education, intelligence, parental education and father's occupational class have the largest influence, was there a threshold value for determining that a variable had a large influence? With the total effect of 0.59, I see that education is a strong predictor, but the other three are 0.16-0.19, and these are closer in magnitude to the other predictors (0.03-0.09).

Reviewers' comments:

Reviewer's Responses to Questions

**Comments to the Author**

1. Is the manuscript technically sound, and do the data support the conclusions?

Reviewer #1: Yes

Reviewer #2: Yes

2. Has the statistical analysis been performed appropriately and rigorously? 

Reviewer #1: Yes

Reviewer #2: Yes

3. Have the authors made all data underlying the findings in their manuscript fully available?

Reviewer #1: No

Reviewer #2: Yes

4. Is the manuscript presented in an intelligible fashion and written in standard English?

Reviewer #1: Yes

Reviewer #2: Yes

5. Review Comments to the Author

**Reviewer #1: ** This is a simple and nice paper. It's greatest strength is the use of Danish longitudinal data. As I understand, this data set has so far not been used to study the topic of status attainment and social mobility. It's good that the authors report zero-order correlations, in addition to the more complex path analyses.

A few comments for improvement:

You measure social class by 5 categories, but you never name these 5 categories. I think you should list the class categories, which were used.

On page 23 you say that you used a reflective measurement model, although a formative model would have been better. Please explain the difference between reflective and formative model, why a formative model would have been better and why weren't you able to use it.

Table 4 is a bit unclear. Which variables in the table are dependent variables and which are predictor? What are the numbers in the table - correlations or path coefficients?

**Reviewer #2: ** Review:

I read this paper with great interest, and believe it has the potential to become a widely referenced source on the important topic of life course influences on adult socio-economic status. I will mention some of the important aspects of the paper that make it likely to become a widely referenced source, followed by a few suggestions that might improve the impact of the study and the clarity of the manuscript.

1. This is a well conducted replication and extension of previous studies: It replicates main consistent findings in the literature, and extends these to a relatively large, new population, context, and age cohort, with a broad range of childhood SES backgrounds. In addition there are important new features here: e.g., the multiple time points for prediction to mid adulthood (early childhood, later childhood, early adulthood); the addition of a range of predictors that have not been widely examined (these typically show small direct effects, but are important mediators of earlier predictors); and, most importantly, the focus on indirect effects and mediation of established predictors.

2. This design allows for comparison of the relative effect sizes (both direct and indirect) of predictors. It will therefore be a useful reference for future studies with other cohorts.

3. Studies directly focused on prediction of mid-adult SES are relatively unusual (often SES is considered as a predictor or control variable in longitudinal designs rather than an outcome). As the effects of the “established” predictors appear to vary somewhat across populations, contexts, etc. it is important to add to the body of literature with additional projects with different populations, contexts, age cohorts etc. so that this variability of results can be interpreted and understood. The importance of replication in a research area which is largely observational is well established.

4. This design allows identification of important “intermediate” outcomes that may be more amenable to intervention and policy decisions than the established predictors (e.g. parental education, occupational status, childhood IQ).

There are some limitations in the present paper based on the data set, which are well identified and stated in the manuscript. Most important of these are the limitations of using a single sex (male) which was available from the original sample, and some limitations in the type of predictive measures which were available from the data set (as described in the paper).

My main concern regarding limitations is that moderation effects were not reported. In addition to mediation, moderation is likely to be a main source of variance in this type of analyses. The authors explain that it would require an entire additional manuscript and analysis to examine these, but I would argue that we only have “half the story” here. It is an extremely interesting and potentially useful “half story” and so will be of broad interest to readers, but from both conceptual and applied/policy perspectives, the moderation processes should be considered as well. Could there be an appendix, or a related/complementary paper made available from the data set?

A minor specific issue: Please explain why weighted loadings were used for occupational status and income (p.9, ln.19). This influenced the results, as stated, but we don’t know what the rationale was….

In sum, I would like to know more about the interaction effects (moderation) in this or a complementary report. However, I am very impressed with the information and design presented here. I think it has the potential to be widely read and replicated, as well as used for social and educational policy development.

6. PLOS authors have the option to publish the peer review history of their article (what does this mean? ). If published, this will include your full peer review and any attached files.

**Do you want your identity to be public for this peer review?** For information about this choice, including consent withdrawal, please see our Privacy Policy .

Reviewer #1: No

Reviewer #2: **Yes: ** Lisa A. Serbin, Professor of Psychology, Concordia University, Montreal, Quebec, Canada

---

## [Author Response · Author response to Decision Letter 1]

1 Jul 2025

Journal Requirements:

1. When submitting your revision, we need you to address these additional requirements. Please ensure that your manuscript meets PLOS ONE's style requirements, including those for file naming. The PLOS ONE style templates can be found at https://journals.plos.org/plosone/s/file?id=wjVg/PLOSOne_formatting_sample_main_body.pdf and https://journals.plos.org/plosone/s/file?id=ba62/PLOSOne_formatting_sample_title_authors_affiliations.pdf

Response: We have previously published in PLOS One, and we believe that our manuscript meets the requirements of the journal.

2. We note that you have indicated that there are restrictions to data sharing for this study. For studies involving human research participant data or other sensitive data, we encourage authors to share deidentified or anonymized data. However, when data cannot be publicly shared for ethical reasons, we allow authors to make their data sets available upon request. For information on unacceptable data access restrictions, please see http://journals.plos.org/plosone/s/data-availability#loc-unacceptable-data-accessrestrictions. Before we proceed with your manuscript, please address the following prompts:

Response: Our original data availability statement was relatively short, providing information on how to proceed to get access to data stored at Statistics Denmark.

Statistics Denmark is a Danish public institution that continuously receives individual-level data on all Danes (e.g. data on income, address, civil status etc.). Because these data are highly sensitive and Danes cannot choose not to be registered, Statistics Denmark has strict rules concerning data access by researchers. Thus, individual researchers can only access data in Statistics Denmark through institutions (e.g. university departments) that have been approved by Statistics Denmark (access can be through a VPN connection on a computer approved by the institution). Furthermore, in Statistics Denmark individuals are identified by a 10-digit personal identification number, but researchers only get access to de-identified databases without the personal identification number.

In the context of our paper, the important points are: The research group has no influence on the data access rules of Statistics Denmark. Data from Statistics Denmark are not allowed to be included in publicly available de-identified databases, but Danish and foreign researchers can get access to de-identified data through an institution approved by Statistics Denmark.

We have revised the Data Availability Statement:

Due to restrictions on access to data stored at Statistics Denmark, raw data from the present study cannot be made available in public de-identified databases, but researchers can get access to de-identified data through an institution approved by Statistics Denmark. Queries regarding data access and more information about the cohort can be found on the homepage: https://www.frederiksberghospital.dk/ckff/sektioner/SBF/psykiatriskepidemiologi/Sider/

The-Danish-Metropolit-cohort.aspx.

Response: We apologize for this error, and the two abstracts should now be identical.

Additional Editor Comments:

In addition to the comments from the 2 reviewers, please consider the following:

1. Please review the PLOS One guidelines for statistical reporting and ensure that the manuscript complies (e.g., exact p values when greater than 0.001, confidence intervals etc): https://www.equatornetwork. org/wp-content/uploads/2013/03/SAMPL-Guidelines-3-13-13.pdf

Response: Table 1 is a descriptive table, but we have added the information that “Correlations at 0.03 or larger are statistically significant”. Because the table is descriptive, we do not find it necessary to include information on the significance level of each correlation.

In tables 2, 3, and 4, significance is indicated by the often-used star system (* p<0.05; ** p<0.01; *** p<0.001). We have used this system in previous PLOS One papers and find it appropriate here because exact p values would require much more space and also because we find that the pattern of results is more important than the exact p values. Finally, we note that the star system is used in several PLOS One papers using SEM models.

Should the editor disagree, the stars can of course be replaced by exact p values.

2. Please also consider guidelines for SEM analysis and ensure that all relevant details are included in the manuscript (e.g., https://doi.org/10.1016/j.anr.2021.06.001). It may help to review how other SEM-based papers in PLOS One have reported their findings. It seems that a path analysis diagram showing the direct and indirect effects is warranted, and may also help illustrate the "trajectories of influence" that the authors mention in their conclusion.

Response: The author group has had long discussions about how to present the structure of our data in a figure. The main problem is the large number of predictors from three age periods. Attempts to construct a traditional SEM figure including all variables and effects showed that such a figure was not feasible (considering the number of direct and indirect paths for 5 early childhood predictors, 9 late childhood predictors and 3 young adult predictors). As an alternative, we produced the simplified figure 1, which shows the temporal structure, the direct effects from each age period, and numbers referring to the description in the data analysis section and the regression models in tables 2-4 (we have made the meaning of the numbers clearer in a revised figure). This figure is rather complex, and for the revised manuscript we have produced a simplified figure 2 showing the indirect paths between early and late childhood predictors and midlife SES. We refer to this figure in the description of indirect effects in the data analysis section.

In general, for a non-SEM expert, I found it difficult to follow the results and in their current form, some of the tables cannot stand alone from the text as it isn't clear what values are presented.

Response: Because the direct effects correspond to standardized coefficients in linear regression models, we assume that it is most difficult to follow the discussion of indirect effects and hope that figure 2 will make it easier to follow both the result section and the discussion of these effects.

Thank you for the observation about the tables. We assume that this was primarily a problem with table 4 – as pointed out by reviewer 1. To accommodate this concern we have expanded the title of table 4: “Standardized Direct Effects on Late Childhood Characteristics Predicted by Early Childhood Characteristics”.

For table 2 we also included the following footnote:”The variables without a coefficient for direct effects were not included in the regression model predicting midlife socioeconomic status”

3. Please provide the rationale for only including father's occupational class.

Response: It is important to keep in mind that the members of the Metropolit cohort were born in 1953 – meaning that the parents most likely were born in the 1920s and 1930s. For these generations of Danish women, the great majority were housewives and not employed in the labor market. This was the reason that only the occupation of the father was coded when the cohort was established and indeed the reason that only boys were included in the cohort. The main focus was to study social mobility and the comparison of mothers’ and daughters’ SES would be difficult even though a substantial part of the 1953 generation of women as adults entered the Danish labor market.

We have added the following sentence to the description of the father’s occupational status “Mother’s occupational class was not registered because the great majority of women were housewives in 1953.”

4. Page 11, Line 16: What preliminary analyses were conducted and where are

these reported?

Response: Our first analyses included variables corresponding to the variables analyzed by Deary et al. (2005), but the Metropolit cohort was established with a focus on social mobility, and we were able to add several other potential predictors of midlife SES. Initially, we considered reporting results for a model similar to the model of Deary et al., but we realized that this would overlap with our full model. Consequently, we only report the full model, but where relevant, we compare our results with the results of Deary et al.

As described in the manuscript, we also tried to fit a model with a formative measurement model for midlife SES.

5. Table 1: In the footnote, the authors mention 3 occupational class variables

but I only see 2 listed in the table.

Response: This has been corrected.

6. Page 5, Line 22: Consider clarifying that you are referring to adult height in this sentence.

Response: Thank you for this suggestion. We have made it clear that we refer to adult height.

7. Page 21, Lines 22-24: Regarding the statement that education, intelligence, parental education and father's occupational class have the largest influence, was there a threshold value for determining that a variable had a large influence? With the total effect of 0.59, I see that education is a strong predictor, but the other three are 0.16-0.19, and these are closer in magnitude to the other predictors (0.03-0.09).

Response: There was no threshold value for determining that a variable had a large influence, and in the conclusion, we use the more appropriate “strongest influence”. However, the pattern of the total effects in table 2 indicates that education is clearly the strongest predictor followed by IQ, parental education, and father’s occupation class. Although the absolute coefficients for the latter three variables are closer to the coefficients of the remaining variables than to education, they are still stronger predictors than the weaker variables.

On page 21, we have changed “largest influence” to “strongest influence” to be consistent with the conclusion. It should also be pointed out that the discussion includes a whole paragraph on the importance of education, and this is underlined in the conclusion (“…educational attainment was a particularly strong predictor of status attainment for this generation of Danes”).

Reviewers' comments:

Reviewer #1:

This is a simple and nice paper. It's greatest strength is the use of Danish longitudinal data. As I understand, this data set has so far not been used to study the topic of status attainment and social mobility. It's good that the authors report zero-order correlations, in addition to the more complex path analyses. A few comments for improvement:

You measure social class by 5 categories, but you never name these 5 categories. I think you should list the class categories, which were used.

Response: In the original manuscript, we refer to a paper with a detailed description of the categories (Christensen et al., 2014). We have now included a short description of the categories: “The individual’s occupational class was categorized into five categories in accordance with previous Danish studies [19]. Category 5: University graduates or top-level management positions. Category 4: Jobs requiring medium-long education or management responsibilities for 11-50 subordinates. Category 3: Jobs requiring short theoretical education or management responsibilities for up to 10 subordinates. Category 2: Lower-level white-collar jobs and skilled manual workers. Category 1: Unskilled workers. Self-employment was also coded according to the number of subordinates, but each category required fewer subordinates than salaried employment.”

On page 23 you say that you used a reflective measurement model, although a formative model would have been better. Please explain the difference between reflective and formative model, why a formative model would have been better and why weren't you able to use it.

Response: In a reflective model, the construct (latent variable) determines the observed variables and scores are assumed to reflect a latent trait such as intelligence or personality traits. In the formative model, the observed variables determine the construct, which may be conceived as a combination of the observed variables. We tried a formative model, but it did not converge, and in fact, identification problems are common for formative models (e.g. Edwards, 2011).

We cannot describe the differences between the two measurement models in detail in the manuscript, but we have added a short sentence to the text in the strengths and limitations paragraph:

“First, many would consider a formative model more appropriate if SES is conceived as a combination of occupational class and income. However, our analyses were based on a reflective measurement model because we were unable to fit a formative model, as it did not converge”

Table 4 is a bit unclear. Which variables in the table are dependent variables and which are predictor? What are the numbers in the table - correlations or path coefficients?

Response: To save space, this table shows the outcome variables (late childhood characteristics to the left) and the results of the predictor variables in separate columns. This is different from tables 2 and 3, but to accommodate the reviewer’s concern we have expanded the title of the table: “Standardized Direct Effects on Late Childhood Characteristics Predicted by Early Childhood Characteristics”

It would take unnecessary space to use the format of tables 2-3 for table 4 – since table 4 only shows direct effects.

Reviewer #2:

I read this paper with great interest, and believe it has the potential to become a widely referenced source on the important topic of life course influences on adult socio-economic status. I will mention some of the important aspects of the paper that make it likely to become a widely referenced source, followed by a few suggestions that might improve the impact of the study and the clarity of the manuscript.

1. This is a well conducted replication and extension of previous studies: It replicates main consistent findings in the literature, and extends these to a relatively large, new population, context, and age cohort, with a broad range of childhood SES backgrounds. In addition there are important new features here: e.g., the multiple time points for prediction to mid adulthood (early childhood, later childhood, early adulthood); the addition of a range of predictors that have not been widely examined (these typically show small direct effects, but are important mediators of earlier predictors); and, most importantly, the focus on indirect effects and mediation of established predictors.

2. This design allows for comparison of the relative effect sizes (both direct and indirect) of predictors. It will therefore be a useful reference for future studies with other cohorts.

3. Studies directly focused on predic

---

## [Decision Letter · Decision Letter 1]

28 Jul 2025

A life course perspective on predictors of midlife socioeconomic status

PONE-D-25-14004R1

Dear Dr. Mortensen,

We’re pleased to inform you that your manuscript has been judged scientifically suitable for publication and will be formally accepted for publication once it meets all outstanding technical requirements.

Kind regards,

Heather Macdonald, Ph.D

Academic Editor

PLOS ONE

Additional Editor Comments (optional):

Thank you for addressing my comments and those of the reviewers.

Reviewers' comments:

Reviewer's Responses to Questions

**Comments to the Author**

1. If the authors have adequately addressed your comments raised in a previous round of review and you feel that this manuscript is now acceptable for publication, you may indicate that here to bypass the “Comments to the Author” section, enter your conflict of interest statement in the “Confidential to Editor” section, and submit your "Accept" recommendation.

Reviewer #1: All comments have been addressed

Reviewer #2: All comments have been addressed

2. Is the manuscript technically sound, and do the data support the conclusions?

Reviewer #1: Yes

Reviewer #2: Yes

3. Has the statistical analysis been performed appropriately and rigorously? 

Reviewer #1: Yes

Reviewer #2: Yes

4. Have the authors made all data underlying the findings in their manuscript fully available?

Reviewer #1: No

Reviewer #2: Yes

5. Is the manuscript presented in an intelligible fashion and written in standard English?

Reviewer #1: Yes

Reviewer #2: Yes

6. Review Comments to the Author

Reviewer #1: All good. I have no further complaints. Nothing much to add. But I have to keep writing to attain the count.

Reviewer #2: The authors have been very responsive to reviewer and editor comments. I believe the paper is ready to publish.

7. PLOS authors have the option to publish the peer review history of their article (what does this mean? ). If published, this will include your full peer review and any attached files.

**Do you want your identity to be public for this peer review?** For information about this choice, including consent withdrawal, please see our Privacy Policy .

Reviewer #1: No

Reviewer #2: **Yes: ** Lisa A. Serbin, Ph.D., Professor of Psychology, Concordia University, Montreal, Quebec, Canada

---

## [Editor Report · Acceptance letter]

PONE-D-25-14004R1

PLOS ONE

Dear Dr. Mortensen,

I'm pleased to inform you that your manuscript has been deemed suitable for publication in PLOS ONE. Congratulations! Your manuscript is now being handed over to our production team.

Kind regards,

on behalf of

Dr. Heather Macdonald

Academic Editor

PLOS ONE